# Position: Neglecting the Sustainability of AI is Fuelling a Global AI Arms Race

**Pedram Bakhtiarifard** [1]   **Pınar Tözün** [2]   **Christian Igel** [1]   **Raghavendra Selvan** [1]

## Abstract

Sustainability encompasses three key facets: economic, environmental, and social. However, the nascent discourse on sustainable artificial intelligence (AI) predominantly focuses on the environmental sustainability of AI, neglecting the economic and social aspects. Achieving truly sustainable AI necessitates addressing the tension between its environmental sustainability, which emphasises mitigating AI's climate impact, and its social sustainability, hinging on equitable access to AI development resources. This push for increased accessibility, however, often overlooks the environmental costs of expanding such resource usage. This position paper argues that reconciling climate awareness and resource awareness is essential to realising truly sustainable AI, and neglecting these factors fuels a global AI arms race. Applying Karl Marx's base-superstructure framework from historical materialism, we analyse how the material conditions are shaping the current AI progress and the discourse surrounding it. Further, we introduce the Climate and Resource Aware Machine Learning (CARAML) framework with actionable recommendations spanning individual, community, industry, government, and global levels to achieve sustainable AI.

## 1. Introduction

The capabilities of artificial intelligence (AI) methods are multifaceted, with the potential to help us as a global society to reach many of the Sustainable Development Goals (SDGs) (Vinuesa et al., 2020). The availability of large-scale datasets and computational resources has been essential in the recent accelerated development of deep learning methods, which is driving most of these AI capabilities (Le-

[1]Department of Computer Science, University of Copenhagen, Denmark [2]Data, Systems, and Robotics Section, IT University of Copenhagen, Denmark. Correspondence to: Raghavendra Selvan <raghav@di.ku.dk>.

*Proceedings of the $43^{rd}$ International Conference on Machine Learning*, Seoul, South Korea. PMLR 306, 2026. Copyright 2026 by the author(s).

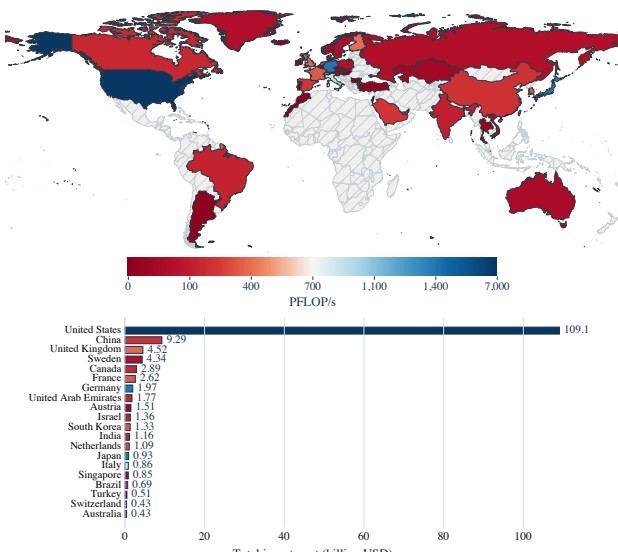

*Figure 1.* Compute capacity **(top)** in different countries based on the Top500 listing of supercomputers per November, 2025 (TOP500, 2025). Top 20 countries **(bottom)** AI investment for 2024 using data derived from Maslej et al. (2025).

Cun et al., 2015; Schmidhuber, 2015). However, the recent advanced AI systems such as large language models (LLMs), and other foundation models behind generative AI are produced by a handful large corporations based in a few countries with access to extraordinary computational resources and infrastructure. These entities invest in thousands of Graphics Processing Units (GPUs), internet-scale datasets, hyper-scale datacenters, and, in some cases, dedicated power grids to develop state-of-the-art generative models (Maslej et al., 2025). Figure 1 **(top)** illustrates the distribution of the world's top 500 supercomputers revealing a stark geographic concentration of these resources.

This disparity aligns with a global investment trend, where the United States, China, and European Union lead in both financial capital and aggregate compute capacity (Figure 1, **bottom**). This pattern persists even among open-source LLMs when drawing on popular public leaderboard data (Moutawwakil & Pierrard, 2023). We observe (Figure 2) that most of the entries belong to corporations that have invested in large-scale compute resources.

AI models are rarely developed in the abstract; they are deeply entangled with commercial imperatives. Their cre-

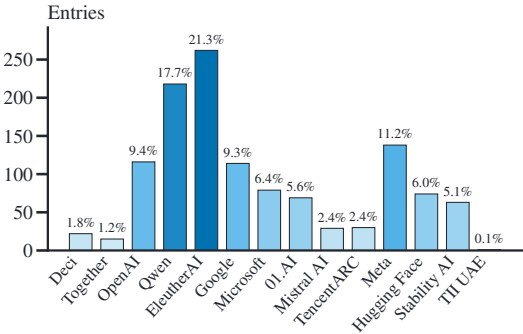

*Figure 2.* Majority of the models listed in the leaderboard for LLM benchmarking on Hugging Face is developed by corporations (Moutawwakil & Pierrard, 2023). This plot shows the number of entries per organization.

ation is aimed, ultimately, at driving up consumption of the companies' own products and services. It was estimated in 2023 that the potential value generated by generative AI models could reach as high as $4.4 trillion (Chui et al., 2023). To put this in perspective, the upper bound of this estimate now exceeds the projected 2026 nominal Gross Domestic Product (GDP) of the United Kingdom, which stands at approximately $4.26 trillion.

Notwithstanding *some* research efforts aimed at using frontier AI systems to accelerate drug discovery (Ocana et al., 2025), improve climate risk forecasting (Camps-Valls et al., 2025), or improve emergency health operations (Sharwood, 2025; Chenais et al., 2023), among others, the immediate societal benefits of these systems remain uncertain. The market-driven push for AI expansion has given rise to a *global AI arms race*, in which public and especially private actors are deploying industrial-scale resources in a bid to secure dominance, while externalising environmental and financial costs, and entrenching social divides. As the computational resources required to develop AI methods have doubled every 3.4 months for (some) salient AI systems (Amodei & Hernandez, 2018; Sevilla et al., 2022) since 2012, it is only reasonable to assume this trend will persist.

According to the International Energy Agency (IEA), datacenter electricity consumption will double from its 2024 level, primarily due to the uptake of AI (International Energy Agency, 2025b). In 2024, private investment in AI in the United States alone reached $109.1 billion, which is nearly 12× that of China, 24× that of the United Kingdom (Maslej et al., 2025), and more than the GDP of 134 countries in 2024. As the AI arms race escalates, countries are bolstering computing capacity, even if the near-term purpose of scaling up is unclear, echoing the nuclear arms race of the cold war. The $52 billion three-year plan from China's Alibaba Group, $11 billion technology innovation fund from the Industrial and Commercial Bank of China (ICBC), the launch of a $8 billion national artificial-intelligence industry fund, and a 20 billion euro in-

vestment in AI giga factories by European Union, all within a single year, exemplifies the deliberate push to expand domestic computing and model-development capacity in the global AI arms race (Lahiri, 2025; Tang & Woo, 2025; Cao, 2025; European Commission, 2025). Alongside this flood of capital investments and resource consumption, a wave of hyperbole that is untethered from principled use of AI has accompanied each new milestone. The hyperbole oscillates between *AI-utopia* and *AI-apocalypse* scenarios, which further skew public and policy perceptions of what AI is and what it is not (Hanna & Bender, 2024).

In this paper, we argue that addressing the unchecked large-scale resource consumption and redirecting the AI discourse away from hype requires a critical perspective rooted in sustainability (Van Wynsberghe, 2021; Selvan, 2025; Wright et al., 2025). Specifically, we present the position that **neglecting the sustainability of AI is fuelling a short-sighted global AI arms race**. This can only be deterred by pursuing sustainable AI which can be achieved by the joint consideration of climate and resource awareness. Climate awareness refers to the recognition of AI's environmental impacts. Resource awareness, in contrast, highlights the infrastructural and economic barriers that restrict participation in AI development, raising urgent concerns around equity, access, and the broader goals of social and economic sustainability. Figure 3 shows how different facets of frontier AI can be contextualised based on the dimensions of climate awareness and resource awareness, spanning state-of-the-art (SOTA) AI, inclusive AI, green AI and sustainable AI. Note that resource awareness should not be confused with pursuing resource efficiency, which is better aligned with *green AI* in Figure 3. Appendix A provides detailed descriptions of each of these AI paradigms within each quadrant.

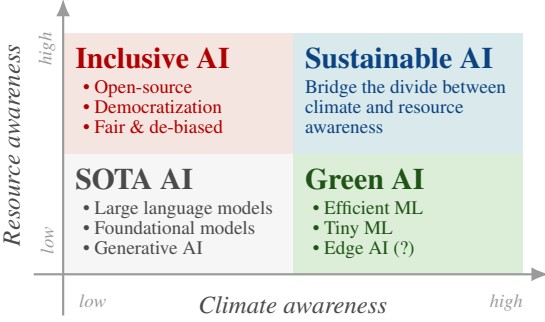

*Figure 3.* Contextualising sustainable AI using the two axes of climate awareness and resource awareness. The position of this paper is that sustainable AI should be in the top right quadrant, where both climate and resource awareness are high. In the case of Edge AI, there is some ambiguity in if it is actually climate aware; hence, we have marked it with a question mark.

Further, we identify a recurring tension between environmental and social sustainability based on the existing literature on sustainable AI. We argue that an exclusive focus on either the climate impact or resource efficiency of AI risks

undermining the broader ambitions of sustainable AI. We draw upon tools from historical materialism, situating contemporary AI development using the *base-superstructure* model developed by Karl Marx. In this view, the material conditions underpinning AI—the computational infrastructure, energy consumption, carbon footprint, capital investment, and labour—form a base that actively shapes and reinforces a superstructure of norms, values, and regulatory frameworks. And such configuration tends to preserve the status quo, at odds with the transformative goals implied of technological change toward more sustainable futures. In response to these challenges, we propose the Climate And Resource Aware ML (CARAML) framework with the focal point that as AI systems grow in complexity, so too must the scope and coordination of the actions taken to preside over their impacts. CARAML offers a set of evolving interventions and recommendations spanning individual, community, industry, governmental and global scales. Each of these is aimed to advance towards more sustainable AI, and deter the global AI arms race.

## 2. Sustainability of AI

Using the lens of sustainability allows us to critically assess the environmental, economic, and social costs of frontier AI. In this section, we will formalise the definitions used in rest of the paper and highlight some contradictions that are manifested when climate awareness and resource awareness are not jointly considered.

### 2.1. AI and the Three Pillars of Sustainability

Sustainable AI is defined in Van Wynsberghe (2021) to concern *"how to develop AI that is compatible with sustaining environmental resources for current and future generations; economic models for societies; and societal values that are fundamental to a given society."* We closely adhere to this notion in this work and discuss the interaction between AI and the three facets of sustainability (environmental, economic, and social). Anchoring these aspects of sustainability and AI is important to advance the position held in this work.

**Environmental Sustainability.** The growing computational demand to develop and deploy frontier AI has led to a proportional increase in the energy consumption (Strubell et al., 2019). As global energy production is still one of the largest greenhouse gas (GHG) emitters (Bruckner et al., 2014), the operational carbon emissions of AI has also been increasing (Anthony et al., 2020; Henderson et al., 2020; Selvan et al., 2022; Luccioni et al., 2023). Furthermore, the manufacturing of specialised hardware required to develop AI models (Falk et al., 2026), construction of the building infrastructure that houses datacenters, the water required to cool them (Li et al., 2025), and the electronic waste generated due to these electronics (Wang et al., 2024) are

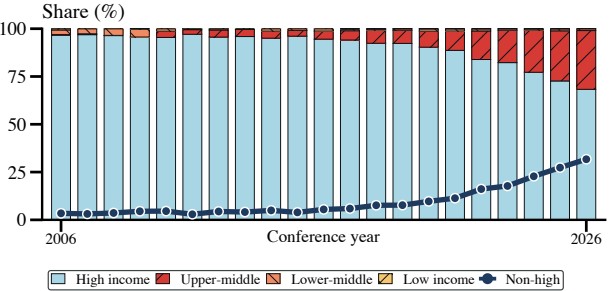

*Figure 4.* Distribution of authors from different income groups based on the country of their affiliations in publications for three premier AI conferences (ICML, ICLR, and NeurIPS) from 2006–2025. Countries are classified based on their income group as defined by World Bank (2024). Data preparation for this plot is detailed in Appendix B.

adversely impacting the *environmental sustainability* of AI methods (Li et al., 2025; Wright et al., 2025; Tannu & Nair, 2023).

**Economic Sustainability.** The rapid expansion of AI technologies has raised pressing questions about their long-term economic viability (Metz et al., 2024). In particular, training and deploying large-scale AI models demands immense financial investments—not just in compute infrastructure, but also in expensive data, skilled labour (Wilson et al., 2026), and continuous maintenance. These costs have led to an increasing consolidation of AI capabilities within a handful of corporations and nations with the capital to sustain such operations. Hence, the economic benefits of AI, such as productivity gains, new market opportunities, and automation, are disproportionately captured by entities that already possess significant resources, thus potentially exacerbating global economic inequality (Hao, 2022; Wilson et al., 2026). This disparity is amplified in historically disadvantaged countries (Salami, 2024; Arora et al., 2023). This is captured in Figure 1 which shows the geographic distribution of the publicly known top 500 super-computers; we observe that the African continent is largely unrepresented.

**Social Sustainability.** AI's large-scale resource requirements have also created a new access barrier to participate in its development and adoption, which increases the risk of *de-democratization of AI* (Ahmed & Wahed, 2020). The resource barrier reduces the participation of stakeholders from several regions of the world, primarily from low- and middle-income countries (LMICs) compared to high-income countries (HICs). For instance, Figure 4 shows the distribution of AI research publications from 2006–2025 based on income groups of countries for three premier AI conferences. We capture a clear under-representation of LMICs, and the distribution has only gotten more skewed in recent years. Disenfranchisement of communities due to high resource costs is also aggravating the already existing biases in AI methods (Ricci Lara et al., 2022; Farnadi et al., 2024).

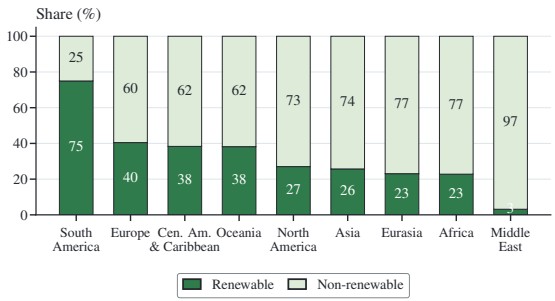

*Figure 5.* Proportion of non-renewable and renewable energy sources in different regions of the world (IRENA, 2024).

### 2.2. Tension between Climate and Resource Awareness

Environmental sustainability is not synonymous with sustainable AI as critically pointed out by Wright et al. (2025). Rather, it is one aspect of sustainable AI pushing for climate awareness, and not concerned with resource awareness, resulting in compute and carbon efficient paradigms such as green AI (Schwartz et al., 2020; Bartoldson et al., 2023). We next present two concrete examples of the contradictions between climate awareness and resource awareness, and their interplay on the overall sustainability of AI.

**Social Sustainability of Low-Carbon Datacenters.** The push towards powering datacenters with renewable energy, or generally known as green computing (Yang et al., 2022), is a valid approach to reduce *some* of the climate impact of developing large AI models—reducing operational carbon emissions. However, green computing solutions, such as low-carbon datacenters, rely heavily on renewable energy like solar, wind, or hydroelectric power to operate. Globally, many countries still face significant challenges in energy access, with large portions of the population lacking reliable electricity or relying on cheaper carbon-intensive sources such as fossil fuels. According to the International Renewable Energy Agency (IRENA), renewable energy capacity in many developing countries (Figure 5) is still insufficient to meet growing demand (IRENA, 2024).

As a result, the demand for clean energy to support greener datacenters for AI not only strains already limited resources (Sukprasert et al., 2024), but also creates a paradox for LMICs. These countries are being asked to adopt greener technologies while lacking the clean energy infrastructure necessary to power them in the first place. This deepens the technological divide and locks out LMICs from participating in the development of AI. This, we argue, results in *double penalty*: once for not having access to the resources, and again for not having access to cleaner (low-carbon) renewables.

**Environmental Sustainability of Efficient AI Models.** The ML community has acknowledged the growing resource costs of developing recent AI models and has primarily

turned towards making them more efficient (Bartoldson et al., 2023). While this in itself is a meaningful endeavour, the larger implications of this resource awareness on the environment are not immediately apparent.

From the perspective of climate awareness, efficiency in computational resource use, though seemingly beneficial in terms of reducing direct energy consumption, can paradoxically be at odds with long-term environmental sustainability when considering rebound effects (Alcott, 2005). This effect, rooted in economic and ecological theory, suggests that any technological advancement that improves efficiency often leads to an increase in overall demand for the very resource it was designed to conserve (Wright et al., 2025; Luccioni et al., 2025).

This paradox had a clear manifestation with the success of DeepSeek models (DeepSeek-AI et al., 2024; 2025), which uses only a fraction of resources compared to other models in the same class.[1] This claim of efficiency has drawn many more people to use and build with these models. That is, these models have improved access, which is a key factor in improving social sustainability. For some companies and countries, DeepSeek may be a sign that "AI-sovereignty" (see Sec. 3) is within reach for them, which may lead to more investments in datacenters and model creation. However, these efficiency improvements in the marginal training and inference costs did not lower aggregate resource consumption. Instead, they triggered a two-tier rebound effect. First, the 2800% surge in DeepSeek's popularity post-release (NIST, 2025; Salim et al., 2026) demonstrates how lower costs drive broader uptake. Second, DeepSeek's efficiency catalysed the "reasoning model" trend, which utilizes "thinking tokens" at orders of magnitude higher volumes than standard LLMs. Current estimates suggest that a typical 80-token response from a reasoning model requires approximately 800 latent thinking tokens (Dauner & Socher, 2025). Thus, even if energy cost per token decreases by 90%, a tenfold increase in token generation results in no net reduction in the total energy use. This shift exemplifies a classic rebound effect where efficiency gains in one area are eclipsed by the overall intensified usage patterns across other areas.

The two cases above have illustrated that solely pursuing climate or resource awareness does not improve the sustainability of AI, and can instead aggravate existing structural inequities or result in rebound effects. So *how* is it we ought to resolve these tensions? Can we steer away from a global AI arms race whilst realising AI sovereignty?

---

[1]We note that even this "smaller" resource cost is exorbitant. DeepSeek-V3 used about $2.8M$ GPU hours on Nvidia-H800s, compared to say $30.8M$ GPU hours on Nvidia-H100s for a comparable model Llama-3-405B.

## 3. The Material Basis for AI

We propose to examine the current frontier AI as a technological transformation shaped by its material base and its societal interactions in order to reconcile the contradictions between climate awareness and resource awareness. We propose to view frontier AI to not be just a hive of algorithms but as infrastructure. We draw upon relevant ideas from historical materialism to situate the current discourse on frontier AI and the ongoing global AI arms race.

**Democratisation of AI.** Ensuring widespread access to, and participation in, the development of any technology is essential for its equitable advancement. This is also true for AI, as it can improve participation in its development (Berditchevskaia et al., 2021) to match the local needs instead of relying on *trickle-down AI* solutions imported from the minority world and HICs. The CEO of NVIDIA, Jensen Huang, captures this essence of *AI sovereignty* by emphasizing that AI codifies a society's culture, intelligence, and history, asserting that nations should "own their own data." (Caulfield, 2024). We agree with this aspiration; however, we are wary of the monopoly held by NVIDIA over global AI infrastructure (Cusumano, 2024). Specifically, we must recognise that the push for AI sovereignty should also include ownership and control of the material infrastructure it is built on.

The pursuit of AI sovereignty, wherein nations develop and control their AI capabilities, stands in contrast to what could be termed *Factional AI*, a scenario where AI development is concentrated among a few powerful entities with disproportionate influence (Ahmed et al., 2023). While AI sovereignty promises a more democratised and inclusive technological future, it is also a driver of the global competition for computational resources, exacerbating existing inequalities (Pasquinelli, 2023). This divide between the *GPU-rich* and *GPU-poor* clearly reflects the contrasting abundance and scarcity mindsets shaping AI development. Countries and corporations with vast computational resources can push the frontiers of AI, while others struggle with limited access to hardware, data, and expertise. In some instances the call to AI sovereignty is also directly fuelling the global AI arms race.

Another consequence of unequal access to the resources needed to develop frontier AI is that it reinforces biases that cater to specific groups while neglecting or misinterpreting others (Wilson et al., 2026). As AI research becomes increasingly dominated by industry interests (Ahmed et al., 2023), the risks of a widening technological divide grow more pronounced, leading to cascading disparities that further marginalize under-represented communities (Farnadi et al., 2024).

However, the pursuit of AI sovereignty and democratised universal access to frontier AI bears the risk of acceler-

ating the ecological degradation unless sustainability considerations are embedded within AI policies and practices. Without careful intervention, unscrupulous AI development may deepen its alienation from nature, individuals, and the broader societal well-being (Marx, 1859).

**AI as Infrastructure.** Training the Llama-3.1 model with 405B trainable parameters on a dataset consisting $> 15$ trillion tokens required 30.84 M GPU hours (AI@Meta, 2024). The corresponding energy consumption of the GPU usage alone can be estimated to be 21.5 GWh, with a total of 8930 tons CO2e in carbon emissions when training in the US. However, geographical variance can significantly alter these impacts. For the same training run, emissions would drop to 750 tons CO2e in a low-intensity region like Sweden, but rise to 14,737 tons CO2e in high-intensity regions like India (Ember, 2025). This is the cost for one *open-source* model; however, exorbitant resources at this or higher scales are being used to develop the wide array of frontier AI models. While one can argue that open-source models can be used by a broader community[2], the majority of the resources being spent are for proprietary models. This is because the material infrastructure required to develop models at this scale is currently available only with a handful of private actors.

In this context, Kate Crawford characterises AI not as an algorithmic advancement but an embodied infrastructure. Crawford writes, *"artificial intelligence is both embodied and material, made from natural resources, fuel, human labor, infrastructures, logistics, histories, and classifications"* (Crawford, 2021). This has been echoed in other works such as Selvan (2025) where the author insists on viewing AI as infrastructure. The unequal access to this embodied infrastructure shapes not only the types of AI models that are created but also how these models are used.

The material basis for AI's development, forming the embodied infrastructure that includes knowledge, labour, investments, and raw materials, does not merely exist in isolation; it also shapes the ethics, policy, cultural narratives, and educational frameworks that govern AI and beyond. The *dominant* ethical frameworks, policy decisions, cultural narratives, and educational systems are all shaped to maintain the concentration of control over the material base. This interplay ensures that AI is deployed in ways that maintain existing power imbalances, rather than challenging them (Pasquinelli, 2023). This framework of viewing a material base that interacts with a superstructure stems from historical materialism, most prominently developed in the work of Karl Marx (Marx, 1859), illustrated in Figure 6, and is particularly relevant in this era of large-scale AI development, given the risks posed by climate change.

---

[2]Meta reported that Llama models have been downloaded $> 350M$ times between Feb. 2023 and Jul. 2024 (Al-Dahle, 2024).

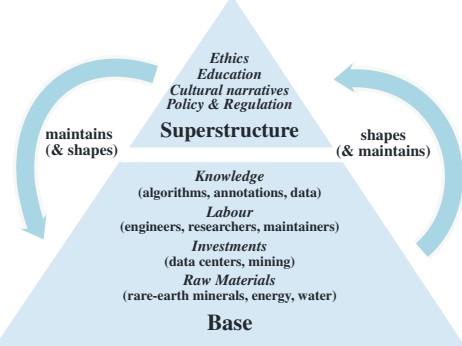

*Figure 6.* Illustration of the base-superstructure framework based on Karl Marx's analyses in Marx (1859). The material basis for developing AI consists of raw materials, monetary investments, labor, and knowledge commodities. This base forms the foundation of AI as an infrastructure which also shapes the socio-political identity of AI—manifested through policy, regulation, cultural narratives, media, and education—which can be considered as the superstructure. The material base shapes the superstructure, which in turn maintains and shapes the base. Neither the base nor the superstructure is static; they influence each other.

**Base-Superstructure Framework for AI.** The base-superstructure framework posits that the economic foundation of society, comprising the forces and relations of production, dictates the legal, political, and ideological superstructure (Marx, 1859). In this model, the base is the primary driver of social change, while the superstructure serves to legitimise and maintain the existing economic order.

Consider applying this to the era of colonial expansion (and discussing this topic in just a single paragraph). The colonial base was built on capital accumulation and forced labour, which necessitated a superstructure of racial hierarchies and administrative control to stabilise resource extraction. This arrangement institutionalised a global division of labour that funnelled wealth to Europe, providing the capital required to fuel the Industrial Revolution. Rather than overturning this system, industrial capitalism utilised existing legal and military frameworks to maintain colonies as captive markets and providers of cheap raw materials. The superstructure of international law and education served to maintain this European hegemony. This feedback loop between the economic base and its justifying institutions ensured that colonial powers remained the global status quo (Harvey, 2017). This is a structural dynamic that continues to shape the distribution of power and resources, well into the ongoing frontier AI revolution.

Although the base-superstructure framing was developed to explain the changing social relations during the Industrial Revolution, it is almost immediately applicable to how modern AI systems are developed in a globalised, market-capitalist context where the means of producing this technology is extremely concentrated (Pasquinelli, 2023). The "Base" now comprises the material resources, data, and specialised knowledge currently monopolised by a few corpora-

tions. This concentration of power in the Global North relies on extractivist mining, data appropriation from the Global South, and the outsourcing of downstream harms (e.g., traumatic content moderation) to the Global South (Wilson et al., 2026). The "superstructure" comprising AI policy, legal frameworks, and research norms is steered to maintain these exploitative relations.

**Illustration of the Base-Superstructure Interplay in AI.**
Consider a *hypothetical* but realistic scenario where an advanced AI hardware producer faces export restrictions due to political and economic tensions (Cheng, 2024). These restrictions, justified under national security and technological leadership narratives, exemplify how the superstructure—through policy, regulation, and strategic discourse—can exert control over the material base, which in turn shapes the trajectory of AI development.

In response, the restricted entity must rapidly reconfigure its supply chains, invest in alternative semiconductor manufacturing, and develop domestic expertise to regain technological independence. However, this forced adaptation is constrained by existing dependencies controlled by the dominant actors forming the superstructure. The initial effect is technological deceleration, increased production costs, and fragmentation of global AI development efforts, reinforcing existing asymmetries rather than dismantling them.

Meanwhile, the controlling superstructure uses these constraints to strengthen its own material dominance. By limiting access to critical AI hardware, it dictates the pace and direction of innovation elsewhere, ensuring that alternative ecosystems develop under more challenging conditions. At the same time, narratives around security, ethical AI governance, and responsible innovation are deployed to justify these material restrictions, further legitimising the existing power structure.

This feedback loop illustrates how the superstructure does not passively reflect material conditions but actively intervenes to maintain and reshape them. Even as new production centers emerge, they do so within a landscape already defined by the controlling actors, reinforcing a cycle where technological dependencies still persist, just under a different guise (Pappachen & Ford, 2023).

In this paper, we asked whether we can steer frontier AI developments away from a global AI arms race whilst still realising AI sovereignty. This, we argue, requires climate and resource awareness to shape the superstructure and adjust its relationship with the material base. But it has to start from the base, since many of the readers are also part of building and maintaining it. An equal call for climate awareness must therefore accompany a call for access to resources; otherwise, resource expansion risks recapitulating the same dynamics that sustain the status quo.

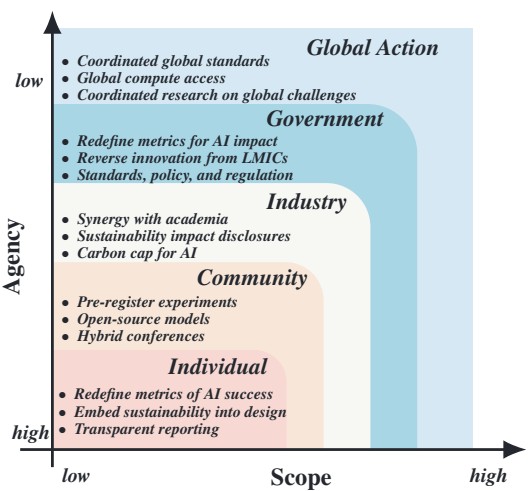

*Figure 7.* Our proposed call to action shown as the CARAML framework. The actions span individual, community, industry, government, and global levels to achieve sustainable AI.

## 4. Call to Action

Deterring the global AI arms race and improving the sustainability of AI requires us to engage actions at multiple levels that work together. If we are to be inspired by the base-superstructure framework to dismantle the global AI arms race, it would require a democratic co-opting of the superstructure and aligning it with sustainable AI. Changing the status quo, historically, has been a high entropy process that has led to many revolutions. In this paper, we do not call for a radical revolution, but we suggest a concrete call to action, nonetheless. We outline some ideas that are already prevalent in the literature, and chalk out other new suggestions, within the CARAML framework that evolves across individual, community, industry, government, and global levels. These recommendations stem from the critical view of AI using the base-superstructure framework that aims to reconcile climate awareness and resource awareness.

**CARAML Framework** Figure 7 illustrates some action points along the axis of agency and scope. We use these axes, similar to the framing by Wright et al. (2025) to illustrate that as individuals we have a lot agency but the scope of our actions can be limited. In order to maximise the sphere of influence, i.e., reach global action, it requires a concerted, multi-stakeholder effort. The specific points mentioned at each level should be seen as suggestions that are tailored to the ML community. For illustration, we mention one point from each level next. More detailed descriptions are presented in Appendix 2.

○ **Individual** → *Redefine Metrics of AI Success.* Performance measures of models should account for sustainability, as focusing only on task-specific measures like accuracy obfuscate climate and resource awareness. Performance that

is normalized to account for resource costs is more useful in this regard (Evchenko et al., 2021; Selvan et al., 2025).

○ **ML Community** → *Pre-Register Experiments.* Large-scale AI experiments should be pre-registered as it enhances transparency and accountability. It can also prevent wasted resources on flawed, repetitive, or misleading research, ensuring investments in AI contribute to long-term, ethical progress rather than short-term hype. Finally, it can help prioritize responsible experiments that justify their resource costs, discouraging redundant or wasteful computations (Albanie et al., 2022).

○ **AI Industry** → *Carbon Cap.* Industry should commit to a regulatory limit on total carbon emissions due to AI. Given the skyrocketing energy demands of large-scale AI training and deployment (Sevilla et al., 2022), this approach would force the industry to prioritize sustainability and accountability.

○ **Governments** → *Redefine Metrics for AI Impact.* Governments could require mandatory AI Impact Assessments (AIAs) similar to Environmental Impact Assessments (EIAs) (Reisman et al., 2018). These would evaluate potential AI impacts on employment, social structures, privacy, and environmental sustainability, ensuring projects are aligned with public welfare.

○ **Global Action** → *Address Digital Divide.* Governments and international organizations should make AI tools, infrastructure, and knowledge accessible to all. This could involve providing affordable cloud-based AI platforms, open-source models, and training programs to build local AI expertise (Sastry et al., 2024). The question of "Right to Compute" is already being raised (Shearer et al., 2024) which needs to be adjusted to the "Right to Sustainable Compute".

**CARAML Principles in Practice.** The open-source and open-weight foundation model movement, including projects such as LLaMA (Dubey et al., 2024), Falcon (Almazrouei et al., 2023), BLOOM (Luccioni et al., 2023), and Mistral (Karamcheti et al., 2021), offers a useful test-bed for the CARAML framework. These models show that access can be widened, methods can be shared, and development can move beyond closed systems. But CARAML asks whether this push towards openness is aligned with climate awareness and resource equity.

At the individual level, developers are shifting success metrics toward efficiency, accessibility, and real-world impact, especially for low-resource contexts. Resource consumption, for example, was part of the LLaMA model card (AI@Meta, 2024), and many ML conferences are encouraging reporting of compute costs (Sevilla et al., 2023). The ML community benefits from increased transparency and open sharing of methods and data. This is currently coalescing around platforms like Hugging Face, but wider adoption of standardised climate impact reporting is still needed (Mitchell et al., 2019).

Within the AI industry, open-sourcing models fosters collaboration with academia and independent researchers, yet training remains concentrated in a few resource-rich institutions. Without commitments to sustainability disclosures and resource limits, open efforts risk replicating the same unsustainable patterns. At the government level, regulatory and funding frameworks lag behind AI's rapid growth, particularly in lower-income regions. Investments in sustainable compute infrastructure and support for innovations from resource-constrained contexts are essential. At the global level, open-source AI lowers barriers and encourages collaboration, but can also lead to fragmented duplication and increased environmental costs. Coordinated global standards for emissions, responsible releases, and equitable compute access are critical to avoid a global AI arms race.

Open-source models are a good starting point toward sustainable AI as they demonstrate what is possible, but realising sustainable AI requires further collective action that go beyond open-source models and transparency efforts.

## 5. Alternative Views

The caution we ask the community to exercise about the growing climate impact of AI has been disputed in some papers. For example, Patterson et al. (2022) claim that the carbon footprint of ML will plateau and then shrink. The paper was seen as a rebuke to works such as the paper by Strubell et al. (2019) that had estimated the carbon footprint of training AI models. While this is true in absolute terms, the overall energy consumption and carbon footprint of AI have continued to soar due to rebound effects (Wright et al., 2025; Luccioni et al., 2025). Many AI corporations are reporting an increase in their carbon emissions due to AI. For example, Google had "a 13% year-over-year increase and a 48% increase" in their 2023 carbon emissions compared to their 2019 target base year. This is primarily attributed to the increases in datacenter energy consumption (Google, 2024), indicating that the carbon footprint is neither plateauing nor shrinking.

Another line of argument that is presented is that the carbon footprint of the information and communication technology sector has largely remained constant due to the efficiency improvements of hardware. For example, the estimates reported by Malmodin et al. (2024) partially support this claim. While efficiency improvements are important, and can certainly reduce the corresponding carbon footprint, it is not enough to achieve sustainable AI as argued by Wright et al. (2025). Furthermore, AI model training has been proclaimed as "carbon-neutral" because datacenters procure renewable energy certificates or offsets. We note that emission reductions through carbon crediting projects seldom meet claimed emission reductions (Probst et al., 2024).

It is sometimes suggested that the rise of smaller, distilled models—LLama (Dubey et al., 2024), QLoRA (Dettmers et al., 2023), or Phi (Abdin et al., 2024) variants—has already addressed AI's sustainability challenge. We contend that this optimism is premature because the initial carbon footprint of the larger models is amortised over the distilled models. Training these models remains confined to a few actors, so the compute divide persists, and lower inference cost does not curb embodied emissions of expanding compute infrastructure.

Finally, it is often noted that improving the sustainability of AI will not, on its own, solve climate change. There is obviously merit to this sentiment. While reducing emissions from AI's increasing energy demand will not by itself resolve climate change, the scale of that demand remains significant. IEA projects that AI could consume approximately 945 TWh of electricity by 2030. This represents $< 5\%$ of global electricity use, yet already exceeds the annual consumption of Japan. At current average carbon intensity levels, supplying this electricity would produce roughly $447M$ tonnes of carbon dioxide equivalent emissions, which is nearly half the total emissions from global commercial aviation in 2023 (International Energy Agency, 2025a). Postponing action until AI's electricity share becomes larger risks entrenching a higher baseline of emissions.

## 6. Conclusion

We are at the precipice of massive changes in our world. Climate change is at our doorstep, and we as a global community are lagging behind in the effort required to meet any reasonable planetary warming targets. These effects have a disproportionate impact on poorer and disadvantaged communities (Hallegatte & Rozenberg, 2017).

And concurrently, we are also on the verge of creating one of the most promising technological capabilities with the recent advancements in AI. The hype and doom around AI is detrimental to a meaningful discourse (Hanna & Bender, 2024). And also viewing AI as a panacea to all problems, including climate change adaptation and climate change mitigation, is naïve (Klein, 2014). However, when faced with a planetary-scale problem, we should have all the tools at our disposal, which also includes AI methods. Except that it has to be with a key difference. We have argued in this paper that it should be sustainable AI that we should strive for. The discourses around democratisation of AI, AI sovereignty, green AI, and inclusive AI are all interrelated. We as a community should resist the forces that pit us against each other in a global AI arms race. We have more important problems that are actually threatening our existence, like climate change, to address. For the AI solutions we develop to have a net positive impact, we have to be aware of both their climate impact and resource costs.

**Generative AI Usage Statement** ChatGPT version 5.2 and Google Gemini were used to support programming tasks, including the development of scripts for visualization, to edit and refine language and grammar in selected sections of the manuscript.

**Acknowledgements** PB and RS acknowledge funding from the European Union's Horizon Europe Research and Innovation Action program under grant agreements No. 101070284, No. 101070408 and No. 101189771. RS also acknowledges funding from the Independent Research Fund Denmark (DFF) under grant agreement No. 4307-00143B. CI acknowledges support from the Danish National Research Foundation (DNRF) through the Pioneer Centre for AI (grant no. P1) and the Center for Remote Sensing and Deep Learning of Global Tree Resources (TreeSense, grant no. DNRF192). PT acknowledges the support of the Novo Nordisk Foundation Natural and Technical Sciences program under grant agreement number NNF22OC0079398. The authors thank members of SAINTS Lab and RAD and DASYA groups for valuable discussions.

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

# A. Positioning of Sustainable AI in the Broader AI Landscape

We scope out Sustainable AI within the broader AI landscape using climate awareness and resource awareness as two orthogonal axes, shown in Figure 3-B.

**State-Of-The-Art (SOTA) AI.** The strides made with recent generative AI models, in particular LLMs, have revolutionized multiple domains beyond natural language processing with popular applications such as ChatGPT built on GPT-3/GPT-4 (Brown et al., 2020; Achiam et al., 2023) and open-source alternatives built on models such as LLaMA (Touvron et al., 2023). Text-to-image models such as diffusion models (Ho et al., 2020; Rombach et al., 2022) and other foundation models (Kirillov et al., 2023; Ma et al., 2024; Bommasani et al., 2021) have also proven to be powerful in modeling complex and diverse data distributions with versatile capabilities.

While the astonishing capabilities of these SOTA models are one common aspect between them, the other commonality is their extreme reliance on large-scale resources (Sevilla et al., 2022; Maslej et al., 2025; International Energy Agency, 2025b). They all require massive amounts of data to train extremely large models (with several billion parameters) for several hundred GPU days with large energy consumption and carbon emissions (Luccioni & Hernandez-Garcia, 2023).

In addition, the centralized development of these models by few entities excludes out other stakeholders. This demonstrates the low resource awareness of SOTA AI methods, as the questions about how to ensure ethical use of these models and measures to mitigate bias are not intrinsic to how they are developed, but are seen as post hoc solutions (Bender et al., 2021; Ricci Lara et al., 2022).

In our assessment, SOTA AI methods have low climate awareness due to their large resource costs, and also low resource awareness due to their exclusionary and centralized development, putting them in the bottom-left quadrant of Figure 3.

**Green AI.** The life-cycle of a typical DL model consists of several stages: dataset curation, model selection, model training, model development, and model deployment. Each of these steps is resource-intensive. Recent works demonstrate the benefit of data, work, and hardware resource sharing for reducing the CPU needs of data loading and preprocessing steps (Xu et al., 2022; Audibert et al., 2023; Mohan et al., 2021; Robroek et al., 2025), and the GPU needs of actual training (Espenshade et al., 2024; Robroek et al., 2024; Strati et al., 2024; Wang et al., 2021). Reducing the reliance of model deployment on large-scale resources is an active research area in the domain of resource-efficient ML (Sze et al., 2017). For example, recent works target the use of existing hardware resources more effectively in data centers (Strati et al., 2025; Yu et al., 2022), thus reducing the need for DL specific hardware (Jouppi et al., 2023). Others investigate deployment on resource-constrained platforms (Bayer et al., 2024; Hojjat et al., 2024), enabling TinyML (Dutta & Bharali, 2021) or EdgeML/EdgeAI (Zhao et al., 2021). There is significant algorithm development to improve resource efficiency in other steps of the DL life-cycle (Schwartz et al., 2020; Bartoldson et al., 2023). This comprises quantization (Dettmers et al., 2021), model compression (Cheng et al., 2018), efficient model selection (Bakhtiarifard et al., 2024), knowledge distillation (Sanh et al., 2019), and a combination of these steps. However, there are no theoretical paradigms that can holistically improve the resource efficiency of the entire DL pipeline.

On the other hand, it has been shown in multiple recent works that the techniques that improve the resource efficiency of AI methods can, beyond a small degradation in downstream performance, hamper other model attributes such as fairness, privacy, and robustness. This affects the social sustainability of these climate-aware AI methods (Hooker et al., 2020; Stoychev & Gunes, 2022; Ramesh et al., 2023). Furthermore, deploying billions of resource-constrained devices for utilizing TinyML/EdgeAI creates both a resource availability challenge and increases the end-to-end environmental footprint of these methods (Prakash et al., 2023).

It can be disputed if improved efficiency corresponds to high climate awareness, however, existing work on Green AI is advertised to be so (Wright et al., 2025). As a result, we position the class of Green AI methods in the lower right quadrant of Figure 3-B with high climate awareness but low resource awareness, as they do not explicitly grapple with responsible and ethical use of AI.

**Inclusive AI.** The rapid and often unregulated development of AI popularized by the "Move fast and break things" paradigm has outpaced the formation of cohesive ethical frameworks, resulting in a fragmented and contested landscape of algorithmic ethics (Chesterman, 2021). This ambiguity complicates efforts to define and ensure the social sustainability of AI, as different stakeholders interpret ethical responsibilities and social impacts in divergent ways. Topics related to the ethical use of AI models (Jobin et al., 2019), ensuring user privacy (Abadi et al., 2016), fairness (Ricci Lara et al., 2022), and bias mitigation strategies (Bender et al., 2021) are considered to be within the purview of the social sustainability of AI. Techniques that are striving to improve these aspects can be labeled as inclusive AI methods.

Measuring and optimizing for factors that affect the inclusive use of AI methods, such as privacy, fairness, and bias, is not straightforward. Consider fairness, for instance, which is difficult to optimize for when performing standard DL (Corbett-Davies et al., 2023). Existing approaches enforce additional optimization criteria which could be at odds with the task-specific performance measures (Li et al., 2023). Furthermore, the computational overhead due to these can reduce the environmental sustainability of AI methods. This is most clearly captured with differentially private DL, which is notorious for the additional computational overheads (Bu et al., 2023). Given these factors, we place Inclusive AI in the upper-left quadrant of Figure 3, indicating the trade-off between resource awareness and climate awareness.

Given these discussions based on the axes of climate awareness and resource awareness, we ideally want Sustainable AI to be in the upper-right quadrant of Figure 3, which jointly advances climate and resource awareness.

## B. Data Collection and Processing

We collected metadata for accepted papers from ICML, NeurIPS, and ICLR using the conference websites and OpenReview. From this metadata we extract paper titles, authors, and affiliations. Since one paper can list multiple affiliations, we use a weighted paper count, where each paper contributes a total weight of one. If a paper has $k$ unique affiliations, each affiliation receives weight $1/k$. We then map each affiliation to a country, and map the country-year pair to the corresponding World Bank income group (World Bank, 2024). Affiliations that cannot be resolved are recorded as *Unknown* and excluded when reporting shares of papers; these are reported over resolved papers. The numbers of papers scraped and excluded are summarised in Table 1.

**Limitations.** Some affiliations are missing, generic, or ambiguous. In addition, geocoding can return the wrong location for globally distributed organizations, so we use deterministic institution overrides and manually reviewed geocoding cache entries where needed. We report the unresolved weighted-paper count for each conference-year.

*Table 1.* Counts by conference and year. We report the number of entries, the number of unique papers (by title), the split by income group (High-income countries (HIC), Upper-middle-income countries, Lower-middle-income countries, and Low-income countries), and the number of *Unknown* affiliations.

| Year | Total | Unique | HIC | UMIC | LMIC | LIC | Unknown |
|---|---|---|---|---|---|---|---|
| **ICML** | | | | | | | |
| 2017 | 981 | 434 | 407.3 | 18.3 | 3.6 | 1.0 | 3.8 |
| 2018 | 1453 | 621 | 573.1 | 38.3 | 5.2 | 0.3 | 4.0 |
| 2019 | 1910 | 773 | 719.1 | 42.6 | 4.9 | 0.5 | 5.9 |
| 2020 | 2902 | 1225 | 979.8 | 81.7 | 15.1 | 1.7 | 146.8 |
| 2021 | 3117 | 1183 | 1059.5 | 100.5 | 14.2 | 0.8 | 7.9 |
| 2022 | 3329 | 1233 | 1029.9 | 159.5 | 11.5 | 1.7 | 30.4 |
| 2023 | 5640 | 1865 | 1574.2 | 237.3 | 11.3 | 0.1 | 42.1 |
| 2024 | 6773 | 2610 | 1960.8 | 493.0 | 23.1 | 0.3 | 132.7 |
| 2025 | 9261 | 3257 | 2304.5 | 797.9 | 40.3 | 1.1 | 113.2 |
| **NeurIPS** | | | | | | | |
| 2006 | 434 | 204 | 195.5 | 0.8 | 4.8 | 1.3 | 1.6 |
| 2007 | 526 | 217 | 208.9 | 1.0 | 5.7 | 0.0 | 1.4 |
| 2008 | 609 | 250 | 239.1 | 0.2 | 8.7 | 0.0 | 2.0 |
| 2009 | 631 | 262 | 249.2 | 0.4 | 10.8 | 0.5 | 1.1 |
| 2010 | 694 | 292 | 276.5 | 9.4 | 3.8 | 0.0 | 2.2 |
| 2011 | 720 | 306 | 294.1 | 7.3 | 1.5 | 0.0 | 3.0 |
| 2012 | 911 | 368 | 349.3 | 13.6 | 2.4 | 0.0 | 2.7 |
| 2013 | 807 | 360 | 344.0 | 13.2 | 1.4 | 0.0 | 1.3 |
| 2014 | 1020 | 411 | 387.8 | 14.9 | 5.2 | 0.0 | 3.0 |
| 2015 | 849 | 403 | 385.7 | 12.1 | 3.6 | 0.0 | 1.7 |
| 2016 | 1257 | 568 | 533.5 | 23.8 | 7.0 | 0.0 | 3.6 |
| 2017 | 1663 | 679 | 633.3 | 35.5 | 5.2 | 1.7 | 3.2 |
| 2018 | 2528 | 1009 | 910.2 | 86.0 | 4.6 | 0.5 | 7.6 |
| 2019 | 3627 | 1427 | 1285.2 | 115.2 | 14.5 | 0.0 | 12.1 |
| 2020 | 5101 | 1996 | 1700.2 | 165.8 | 22.4 | 0.0 | 107.6 |
| 2021 | 7068 | 2467 | 2077.0 | 277.3 | 28.4 | 0.2 | 84.1 |
| 2022 | 9361 | 3055 | 2403.9 | 499.7 | 40.6 | 0.0 | 110.9 |
| 2023 | 6913 | 3218 | 2294.5 | 522.0 | 18.3 | 1.2 | 382.0 |
| 2024 | 9576 | 4035 | 2673.6 | 924.9 | 23.5 | 0.8 | 412.2 |
| 2025 | 10357 | 5286 | 2950.9 | 1248.6 | 41.4 | 2.5 | 1042.6 |
| **ICLR** | | | | | | | |
| 2013 | 23 | 23 | 0.0 | 0.0 | 0.0 | 0.0 | 23.0 |
| 2017 | 198 | 198 | 0.0 | 0.0 | 0.0 | 0.0 | 198.0 |
| 2018 | 873 | 338 | 320.8 | 11.2 | 2.6 | 0.0 | 3.3 |
| 2019 | 1339 | 501 | 470.7 | 26.2 | 2.5 | 0.0 | 1.7 |
| 2021 | 2465 | 862 | 776.7 | 68.7 | 10.0 | 0.0 | 6.7 |
| 2022 | 5975 | 2619 | 1792.8 | 271.6 | 17.7 | 1.5 | 535.5 |
| 2023 | 9785 | 3804 | 2863.1 | 623.7 | 36.9 | 0.0 | 280.3 |
| 2024 | 6557 | 2260 | 1726.3 | 399.7 | 11.9 | 0.2 | 122.0 |
| 2025 | 11076 | 3703 | 2645.5 | 805.1 | 32.7 | 1.2 | 218.4 |
| 2026 | 12957 | 5352 | 3202.4 | 1441.4 | 44.8 | 0.2 | 663.3 |

*Table 2.* Overview of the CARAML Framework with recommendations at multiple levels with the objective of reconciling climate awareness and resource awareness to advance Sustainable AI.

| Level | Recommendation | Description |
| --- | --- | --- |
| **Individual** | *Redefine Metrics of AI Success* | Performance measures of models should account for sustainability, as focusing only on task-specific measures like accuracy obfuscate climate and resource awareness, e.g., using performance normalized to account for resource costs (Evchenko et al., 2021; Selvan et al., 2025). |
| | *Pareto Sustainable AI* | Multi-objective optimization that actively strives to balance performance, climate awareness and resource awareness will be important in pursuing Sustainable AI. Pareto optimization offers tools for obtaining sets of solutions with varying trade-offs (Miettinen, 1999). |
| | *Open-Source* | Publishing trained models and datasets to foster reproducibility can go a long way in improving how we conduct AI sustainably. This can reduce repeated investment of resources to redo experiments and help improve access to expensive models (Eiras et al., 2024). |
| **ML Community** | *Pre-Register Experiments* | Large-scale AI experiments should be pre-registered as it enhances transparency and accountability. It can also prevent wasted resources on flawed or misleading research, and it can also help prioritize experiments that justify their resource costs, discouraging redundant or wasteful computations (Albanie et al., 2022). |
| | *Transparency & Self-reporting* | Broader impact statements in conferences should be extended to include comprehensive sustainability impact reports. Providing tools to measure this impact can help simplify and standardize reporting resource costs using existing carbon tracking tools (Lacoste et al., 2019; Henderson et al., 2020; Anthony et al., 2020), and model cards (Mitchell et al., 2019; Strubell et al., 2019). |
| | *Hybrid Conferences & Satellite Events* | Hybrid conferences and satellite events can enhance accessibility by allowing researchers from diverse backgrounds, including those with financial, geographic, or mobility constraints, to participate. They reduce travel costs and overall carbon footprint due to conference air travel while broadening the audience (Epp et al., 2023; Editorial@Nature, 2024). |
| **AI Industry** | *Synergy with Academia* | Collaborations between industry and academia can help address AI's societal challenges – such as bias, fairness, and inclusivity – by combining resources from industry (data and other infrastructure). It can also help avoid redundant efforts, enabling efficient and responsible resource allocation (Ahmed et al., 2023). |
| | *Sustainability Disclosures* | Estimating the embodied emissions of AI infrastructure is next to impossible due to the opacity in how much of the industry conducts business (Luccioni et al., 2023). Mandating these disclosures in standardized reporting can help assess the actual sustainability impact of the AI industry (Luccioni & Hernandez-Garcia, 2023; Wright et al., 2025). |
| | *Carbon Cap* | Industry should commit to a regulatory limit on total carbon emissions due to AI. Given the skyrocketing energy demands of large-scale AI training and deployment (Sevilla et al., 2022), this approach would force the industry to prioritize sustainability and accountability. |
| **Governments** | *Redefining Metrics for AI Impact* | Governments should mandate AI Impact Assessments similar to Environmental Impact Assessments (EIAs) (Reisman et al., 2018). These would evaluate potential AI impacts on employment, social structures, privacy, and environmental sustainability, ensuring projects are aligned with public welfare. |
| | *Reverse Innovation from LMICs* | Implement programs that not only transfer AI technologies to LMICs but also create mechanisms for reverse innovation, where solutions developed in LMICs are brought back to HICs. This model ensures that AI development is truly inclusive and reflects diverse challenges (Ahmed et al., 2017). |
| | *Ethics, Policy and Governance* | AI strategies reflect national contexts, prompting moves toward regional sovereignty. However, the fundamental challenges of sustainable AI are global. Governance frameworks must address both shared concerns and local particularities to be effective (Mügge, 2024). |
| **Global Action** | *Coordinated Global Standards* | Governments, in collaboration with international organizations should establish global standards for AI ethics, safety, and performance. These standards would provide guidelines for ensuring Sustainable AI. Efforts like the EU-AI Act (European Parliament and Council of the European Union, 2024) are initial attempts at this. |
| | *Address Digital Divide* | Governments should steer policies to improve AI access. This could involve providing affordable cloud-based AI platforms, open-source models, and training programs to build local AI expertise (Sastry et al., 2024). The question of "Right to Compute" (Shearer et al., 2024) needs to become the "Right to Sustainable Compute". |
| | *Coordinated Global Research* | Form international consortia uniting global AI stakeholders to tackle shared challenges like climate change and pandemics. Collaborative efforts can pool resources to develop AI solutions with truly global impact (Kaack et al., 2022). This can be the antidote to rhetoric that encourages a global AI arms race. |

