# OpenReview forum: "Position: Neglecting the Sustainability of AI is Fuelling a Global AI Arms Race"
_ICML.cc/2026/Position_Paper_Track — ICML 2026 Position Paper Track regular_

### Official Review · Reviewer_FLFP · 2026-03-07

**Significance:** 3
**Argument Clarity:** 1
**Rating:** 4
**Confidence:** 3

**Questions:**

1. Could you clarify the paragraph 'Environmental Sustainability of Efficient AI Models'? Specifically, around line 200, the paper argues that efficient models such as DeepSeek may unlock higher investments in countries that see state-of-the-art AI within reach with fewer resources. Is this a negative aspect in your view? Wouldn't more efficient AI systems actually help democratizing AI, by lowering the access threshold in economic terms?
2. What do you mean with 'pre-registering' experiments?
4. How do you see higher-level parts of your call to action actually take place? Especially as governments seem to be increasingly shifting towards AI as a national security asset rather than a shared global effort.

**Alternative Views Section:**

Yes

**Compliance With Llm Reviewing Policy A Conservative:**

Affirmed.

**Discussion Potential:**

3

**Final Justification:**

Authors' response addressed most of my concerns. My final score is still borderline mainly due to the writing style, and some lingering skepticism about the potential for real-world impact of the proposed objectives. However, as the authors did at least partially commit to removing redundancies and polishing their writing in the final revision, I am now leaning towards acceptance.

**Paper Summary:**

The paper is centered around the problem of developing truly sustainable AI. The authors embrace a broader definition of sustainability, that encompasses economic, enviromental and societal factors, and argue that current efforts in sustainable AI mostly fail to holistically address these different but interconnected issues, effectively contributing to the AI arms race that they aim to contrast. After providing an interpretation of the current AI ecosystem in terms of base-superstructure interaction, inspired by Marx, the authors call the 'base' (i.e. the broader AI community) to individual and community-level actions designed to slow down the AI arms race and improve overall sustainability.

**Position:**

Yes

**Position In Title:**

Yes

**Related Work:**

3

**Strengths And Weaknesses:**

**Strengths:**

- The topic of AI sustainability is very timely, and the unitary approach chosen by the authors (though not introduced by the present paper) has the potential to spark discussion in the community, as opposed to more traditional definitions focused solely on environmental impact.
- The position is stated very clearly and repeatedly, making the core idea easily accessible to any reader.
- The authors provide some actionable suggestions for the community, starting from the individual actions that researchers can adopt in their daily practice, up to government-level policy decisions.

**Weaknesses:**

- The evidence brought by the authors does not always support the thesis. For instance, while it is evident that there is a divide between high and low-income countries in terms of AI research, Figure 3 does not clearly convey the data. The volume of papers in top AI conferences in the last few years has increased so much that it would be impossible to see any change in historically scarcely represented locations, compared to the USA. It would probably be better to show these data in terms of fraction of papers coming from a certain geographic area instead of raw numbers. Moreover, it is unclear if the research and industry divide between rich and developing regions is specific to AI and to what extent the same inequality can be observed in other tech fields, such as engineering or physics.
- The paper feels slightly repetitive in its content, as the authors repeat multiple times their position regarding the de-democratization of AI and the arms race it is fueling. As acknowledged before, this point hides some positive aspects, as it contributes in making the main message loud and clear. However, it also results in making some sections of the paper less engaging and pleasant to read.
- In general, the writing may need some further polishing, and some passages look rather obscure. These include some reported in the Questions section.
- (minor) The caption of Figure 7 appears incomplete.

**Support:**

2

---

> ### Author Rebuttal · Authors · 2026-03-30
>
> We appreciate the reviewer’s assessment of our work as "very timely" and their constructive critique. Below, we address the primary concerns by clarifying our existing text and providing new evidence. We also encourage the reviewer to reference our responses to other reviewers for additional context.
>
> ---
>
> ### Q1: Environmental Sustainability of Efficient AI Models:
>
> We utilise efficient models like DeepSeek to highlight the inherent tension between environmental and social sustainability, which is the core conflict being addressed in this position. Our manuscript notes that claims of efficiency often drive increased adoption. This observation refers to the **rebound effect** detailed in our submission (L187-196). To clarify this for the final version, we provide the following expanded evidence:
>
> _While DeepSeek reduced marginal training and inference costs, these efficiencies did not lower aggregate resource consumption. Instead, they triggered a two-tier rebound effect. First, the 2800% surge in DeepSeek’s popularity post-release [3,4] demonstrates how lower costs drive broader uptake. Second, DeepSeek’s efficiency catalysed the "reasoning model" trend, which utilises "thinking tokens" at orders of magnitude higher volumes than standard LLMs. Current estimates suggest that a typical 80-token response from a reasoning model requires approximately 800 latent thinking tokens [1]. Thus, even if energy costs per token decrease by 90%, a hundred-fold increase in token generation results in a net ten-fold increase in total energy use. This shift exemplifies a classic rebound effect where efficiency gains in one area are eclipsed by intensified usage patterns across the industry._
>
> We hope this also addresses the reviewer's concern about evidence not always supporting the hypothesis.
>
> ---
> ### Q2: Pre-registering Experiments
>
> We direct the reviewer to lines 353–362 and the precedent set by the 2021 NeurIPS Workshop on Pre-registration in ML referenced in our submission [2]. For large-scale AI development, this framework involves:
> * Formulating a primary research question.
> * Submitting a formal proposal prior to confirmatory testing.
> * Executing experiments and reporting results only after proposal acceptance.
>
> This structured approach is directly applicable to the development of new foundational models, ensuring that massive computational resources are allocated based on theoretical merit rather than post-hoc justification. Major venues like ICML could institute formal compute budget limitations; results requiring resources beyond these thresholds could then be reviewed under pre-registration guidelines. Under this model, only those proposals deemed valuable by the community would proceed to large-scale experimentation. Because the environmental impact of AI development is ultimately absorbed by the global commons, such a mechanism provides a tangible path to manage unchecked resource consumption and mitigate the current AI arms race.
>
> ---
> ### Q3: Global Cooperation
>
> The reviewer correctly identifies the challenges of global cooperation. We acknowledge that in the current geopolitical climate, where AI is increasingly militarised, the ideals of sustainable AI and the broader ML research often feel decoupled from reality. The CARAML framework advocates for cooperation through the lens of sustainability, necessitated by government-level policy and international law. However, we concede that if foundational international agreements such as climate accords and human rights laws are bypassed, the "Global Action" component of CARAML remains an aspirational target rather than a guaranteed outcome.
>
> ---
> #### Writing Clarity
>
> We are committed to refining the manuscript to eliminate repetitive content and ensure the prose is as focused and rigorous. However, we also point to the comment by R32pM:
>
> >The writing is exceptionally clear and accessible to anyone in the ML audience, but also to people familiar with social sciences.
>
> ### Divide between HICs and LMICs in AI research
>
> We respectfully disagree that Fig. 3 does not convey the disparity in agenda-setting for the AI community between HICs and LMICs. However, we agree that it would be more appropriate to report a normalized version, e.g., a fraction of papers from various geographic groups, which we will include in the revision.
>
> We do point out (L55–L108) that AI is characterized by a particularly strong concentration of compute, infrastructure, and capital, and agree that although such inequality is not unique to AI, it is especially pronounced and consequential here.
>
> ---
> ### References
>
> [1] Dauner, Maximilian, and Gudrun Socher. "Energy costs of communicating with AI." 2025
>
> [2] Albanie, S. et al. NeurIPS 2021 Workshop on Pre-registration in Machine Learning. 2021.
>
> [3] CAISI Evaluation of DeepSeek AI Models Finds Shortcomings and Risks
>
> [4] Salim, Mohamad, et al. "Tokenomics: Quantifying Where Tokens Are Used in Agentic Software Engineering." 2026

---

> > ### Author Rebuttal · Reviewer_FLFP · 2026-04-03
> >
> > Thank you for your thorough response to my concerns. Most of the issues have been addressed, even though I still maintain minor concerns regarding writing style and the potential for real-world impact of the proposals. Hoping that at least the former can be solved in the final revision, I am adjusting my score.

---

### Official Review · Reviewer_32pM · 2026-03-11

**Significance:** 4
**Argument Clarity:** 3
**Rating:** 5
**Confidence:** 4

**Questions:**

- In cases where expanding AI access to low income countries increase carbon emissions in the short term, how does CARAML propose to resolve conflicts?
- What other tools beyond the superstructure framework could be used to analyse the "global AI arms race"?
- Why does the publication inequality data end in 2021? This is odd.

**Alternative Views Section:**

Yes

**Compliance With Llm Reviewing Policy A Conservative:**

Affirmed.

**Discussion Potential:**

4

**Final Justification:**

The rebuttal has addressed my concerns, I have increased my rating accordingly. I hope the promised discussions are included in the final paper.

**Paper Summary:**

This position paper argues that the discourse on sustainable AI, which is emerging as frontier LLMs and VLMs are consuming an unprecendent amount of resources for training and deployment, has been solely focused on environmental sustainability, thereby neglecting the economic and social dimensions of sustainability. The paper identify, from a marxist perspective, two guiding principles that should be considered to achieve truly sustainable AI. The first, climate awareness, which is concerned with AI's environmental impacts including energy and water usage, carbon emissions, e-waste, etc. The second, named resource awareness, is concerned with the economic and infrastructure obstacles that hinder equitable and fair participation in AI development, particularly in the Global South and non-high-income countries around the world.

For this analysis, the paper builds upon historical materialism, in particular, from Marx's base-superstructure framework. From this perspective, the authors analyse how the material conditions that control AI (computational infrastructure, capital, energy, labor), shape and reinforce the narratives, policies, norms and assymetries around AI development. These material conditions, according to the paper, reinforce inequalities and power imbalances. It follows, according to the authors, that considering only the environmental sustainability aspect of AI while ignoring the resource aspect of its development, is fueling a global AI arms race.

The paper goes beyond pure analysis and propose a framework named CARAML (Climate and Resource Aware Machine Learning), which identifies five dimensions (individual researchers, the ML community, the AI industry, goverments and global entities) for which to suggest actionable recommendations, ranging from more transparent AI development, to carbon caps or global regulation. The CARAML framework aims to reconcile resource and environmental sustainability in a single framework.

**Position:**

Yes

**Position In Title:**

Yes

**Related Work:**

3

**Strengths And Weaknesses:**

Strenghts:
- The paper introduces a conceptual contribution (Eg framing sustainability in two orthogonal axis of climate and resource awareness) is novel and useful, and it is communicated very well in figure 4.
- The paper is timely and significant, and is very up to date with recent developments (DeepSeek, IEA). The paper convinces the reader that the problems therein described are important and urgent to address.
- There is some empirical data (Figures 1-3) supporting the argument of global inequality in AI development. While perhaps unsurpising, the geographic concentration of both compute and investment is very clearly stated in this paper.
- Applying Marx's superstructure framework to AI development is really interesting and novel and adds a political theory depth towards AI analysis that is not easy to see in ML position papers.
- The proposed CARAML framework is very well organised across relevant dimensions, and the suggested ideas are solid albeit perhaps not particularly original.
- The alternative views section is solid and actively engages in counterarguments, which is nice to see.
- The writing is exceptionally clear and accessible to anyone in the ML audience, but also to people familiar with social sciences. At some points, it reads like a well-written journalistic piece.


Weaknesses:
- The superstructure framework is applied weakly and very loosely. This framework was developed for analysing class relations in industrial capitalism, and the paper fails to convince that this is the right theoretical tool for analysing AI developments. Stronger arguments and more examples would be welcome.
- The proposed dimension of "resource awareness" is conflicting at times, sometimes mixing resource efficiency (which is also a part of climate awarness) and access equity. While related, these are different problems, and perhaps a different dimension definition would have been more clear and strenghen the analysis.
- Related to the point above, the CARAML framework correctly identifies the tension between climate and resource awareness but provides no mechanisms to resolve conflicts when these two dimensions clash.
- The Jevons paradox applied to DeepSeek is intuitive but it's not fully demonstrated empirically.
- Carbon emissions reporting should include uncertainty ranges for increased credibility.
- Many of the suggestions in CARAML are not novel (open source models, hybrid conferences, sustainability reporting, etc.) and others are purely aspirational and abstract (coordinated global standards, etc). Therefore the framework reads as a useful analysis tool but the contributions in terms of policy recommendations are weak.
- Several figures reeferences in the appendix are broken.

**Support:**

3

---

> ### Author Rebuttal · Authors · 2026-03-30
>
> We thank the reviewer for their critical evaluation and for characterizing our use of the Base-Superstructure framework as "really interesting and novel." Below, we address the specific concerns regarding the application of this framework and the empirical evidence for our claims.
>
> ---
>
> ### Justification for using the Base-Superstructure in AI
>
> Our reading of the _Base-Superstructure_ framework (and broadly Marxist literature) is not that it was only focused on class relations during the Industrial Revolution, but it was dealing with changing social relations under the buttress of new technologies that require large-scale resources and means of productions. This viewing immediately makes it apparent that we are in not a very different setting. In our globalised, market-capitalist context, this framework is uniquely suited to analyse frontier AI development as a manifestation of late-stage capitalism [2]. We will clarify this further in the paper:
>
> _Although the Base-Superstructure framing was developed to explain the changing social relations during Industrial revolution, it is almost immediately applicable to how modern AI systems are developed in a globalised, market-capitalist context where the means of producing this technology is extremely concentrated [2]. The "Base" now comprises the material resources, data, and specialised knowledge currently monopolised by a few corporations. This concentration of power in the global North relies on extractivist mining, data appropriation from the "majority world," and the outsourcing of downstream harms (e.g., traumatic content moderation) to the global South [7]. The "Superstructure" comprising AI policy, legal frameworks, and research norms is steered to maintain these exploitative relations. Together, the base-superstructure and the CARAML frameworks serve as a strategic intervention to subvert this cycle, using global cooperation to reorient the Superstructure toward holistic sustainability._
>
> ---
> ### Empirical Evidence: Jevons Paradox in DeepSeek
>
> We will elaborate further the rebound effect due to DeepSeek like models. **We refer to the response to Q1 of Reviewer FLFP.**
>
> ---
> ### Clarification of Resource Awareness
>
> We acknowledge the potential for overlap between climate awareness and resource awareness. As defined in our text (L87), resource awareness specifically refers to the infrastructural and economic barriers that restrict participation in AI development, distinct from the environmental metrics of climate awareness. We will clarify that resource awareness is distinct from resource efficiency, using Fig. 4 as resource efficiency would fit well into the Green AI quadrant.
>
> ---
> ### Resolving Conflicts via CARAML
>
> The "CARAML in Practice" section addresses these tensions through open-source models and transparent reporting. Resolving the conflict between climate and resource goals requires balancing all three pillars (environmental, social, and economic). Higher-level interventions such as policy-driven "reverse innovation" and global cooperation are essential for navigating these trade-offs. We provide a detailed taxonomy of these efforts in Table 1 (Appendix).
>
> ---
> ### Carbon Emission Uncertainty Ranges
>
> We initially reported emissions for Llama 3.2 training based on the Llama model card [5], which used the US carbon intensity (0.429 kgCO2e/kWh) as the models were trained in that region. However, geographical variance can significantly alter these impacts. For the same training run, emissions would drop to 750 tCO2e in a low-intensity region like Sweden, but rise to 14,737 tCO2e in high-intensity regions like India [6]. If the reviewer was concerned about another carbon footprint estimation, we are happy to clarify further.
>
> ---
> ### Novelty of CARAML framework
>
> We acknowledge that many elements in CARAML framework are derived from literature (L301-306). While many individual components of CARAML have been discussed previously, they have not been synthesized to address the specific tension between climate and resource awareness. By integrating these via the Base-Superstructure lens, CARAML offers a novel mechanism to mitigate the global AI arms race and prioritize long-term sustainability.
>
> ---
>
> ### References
>
> [1] Dauner, Maximilian, and Gudrun Socher. "Energy costs of communicating with AI." 2025
>
> [2] Pasquinelli, Matteo. The eye of the master: A social history of artificial intelligence. 2023
>
> [3] CAISI Evaluation of DeepSeek AI Models Finds Shortcomings and Risks. 2025
>
> [4] Salim, Mohamad, et al. "Tokenomics: Quantifying Where Tokens Are Used in Agentic Software Engineering." (2026).
>
> [5] AI@Meta. Llama 3 model card, 2024.
>
> [6] Ember (2025); Energy Institute - Statistical Review of World Energy (2025)
>
> [7] Wilson, Sophia N., et al. "How Hyper-Datafication Impacts the Sustainability Costs in Frontier AI." (2026).

---

> > ### Author Rebuttal · Reviewer_32pM · 2026-04-03
> >
> > My concerns have been resolved and I consider upgrading my score accordingly.

---

### Official Review · Reviewer_ubS3 · 2026-03-13

**Significance:** 3
**Argument Clarity:** 3
**Rating:** 4
**Confidence:** 4

**Questions:**

1. The manuscript proposes the CARAML framework conceptually. Could the authors include a small empirical demonstration or experiment (e.g., an efficient, low-carbon AI model example) to illustrate how the framework can be implemented in practice

2. Some arguments appear repeated across sections. Could the authors streamline these parts?

3. It would also be interesting to discuss who defines sustainability in the context of AI development. Different actors (e.g., governments, industry, international organizations, or communities) may have different definitions, which could influence social, economic, and environmental outcomes.

4. The manuscript could also benefit from engaging with more literature on AI sustainability, including work on the environmental costs of AI, green AI, and responsible AI development [1,2].

[1] Shi, Meilin, Krzysztof Janowicz, Judith Verstegen, Kitty Currier, Nina Wiedemann, Gengchen Mai, Ivan Majic, Zilong Liu, and Rui Zhu. "Geography for AI sustainability and sustainability for GeoAI." Cartography and Geographic Information Science 52, no. 4 (2025): 331-349.

[2] Hosseini, Mohammad, Peng Gao, and Carolina Vivas-Valencia. "A social-environmental impact perspective of generative artificial intelligence." Environmental Science and Ecotechnology 23 (2025): 100520.

**Alternative Views Section:**

Yes

**Compliance With Llm Reviewing Policy A Conservative:**

Affirmed.

**Discussion Potential:**

3

**Paper Summary:**

This position paper argues that neglecting sustainability considerations in AI development is fueling a global AI arms race. The authors highlight a central tension between climate awareness (reducing the environmental impact of AI) and resource awareness (ensuring equitable access to AI infrastructure). Using a base–superstructure framework, the paper explains how the concentration of compute resources and capital shapes global AI development and reinforces inequalities. The main contribution is the proposed CARAML (Climate and Resource Aware Machine Learning) framework, which outlines multi-level actions—across individuals, research communities, industry, governments, and global institutions—to guide AI development toward greater sustainability and equity.

**Position:**

Yes

**Position In Title:**

Yes

**Related Work:**

2

**Strengths And Weaknesses:**

Strengths

1. The manuscript is promising in its scope, as it attempts to connect AI development with environmental, economic, and social sustainability, which are often discussed separately. The authors use many concrete examples and data points (e.g., compute concentration, datacenter energy use, investment trends) to illustrate how current AI development may affect social inequality, economic concentration, and environmental impacts. This helps make the argument more grounded.

2. The proposed CARAML framework is a constructive element of the paper, outlining possible actions across multiple levels (individual, community, industry, government, and global). The topic is relevant to the ICML community and is likely to encourage discussion about responsible and sustainable AI development.

Weaknesses

1. After proposing the CARAML framework, the paper remains mostly conceptual. It would be much stronger if the authors included a more concrete demonstration, such as a small experimental case study showing how a model or workflow could balance efficiency, accessibility, and low carbon cost in practice. That would help readers better understand what sustainable AI could look like beyond the conceptual level.

2. Some ideas are repeated multiple times across the manuscript, which weakens the overall sharpness of the argument.

3. More discussion of the complex relationships under this topic would strengthen the argument.  For example, improving efficiency may reduce per-model cost but increase overall demand for compute (rebound effects); expanding access to AI infrastructure may improve equity but also increase total environmental burden. These trade-offs could be analyzed more explicitly.

4. Another important question that is worth discussing is who defines “sustainability”. Different actors (governments, corporations, researchers, or global institutions) may define sustainability differently, which could influence how environmental, social, and economic priorities are balanced. Examining this governance dimension would add another layer of depth to the discussion.

5. This paper also misses some important citations which discuss very similar topics like [1] and [2]


[1] Shi, Meilin, Krzysztof Janowicz, Judith Verstegen, Kitty Currier, Nina Wiedemann, Gengchen Mai, Ivan Majic, Zilong Liu, and Rui Zhu. "Geography for AI sustainability and sustainability for GeoAI." Cartography and Geographic Information Science 52, no. 4 (2025): 331-349.

[2] Hosseini, Mohammad, Peng Gao, and Carolina Vivas-Valencia. "A social-environmental impact perspective of generative artificial intelligence." Environmental Science and Ecotechnology 23 (2025): 100520.

**Support:**

3

---

> ### Author Rebuttal · Authors · 2026-03-30
>
> We thank the reviewer for finding our manuscript "promising in its scope" and "relevant to the ICML community," and for recognising its potential to foster critical discussions on sustainable AI. We address the main concerns raised by the reviewer below.
>
> ---
>
> ### CARAML in Action
>
> While we utilise several case studies to motivate our position (as also pointed out by the reviewer), we agree that an end-to-end example would further clarify the CARAML framework's operational utility. We will include the following use-case to climate change related disaster forecasting in the Bengal Delta in South-East Asia:
>
> _**Use Case Demonstrating CARAML in Action:**_
>
> _A purely technosolutionist approach treats the prediction of floods in the Bengal Delta as a data problem to be solved with more parameters and higher compute. In contrast, the CARAML framework views this task through the **Base-Superstructure** lens, recognising that "solutions" often carry hidden material costs (Base) and reinforce global power imbalances (Superstructure)._
>
> _At the level of **individual researchers and the ML community**, the focus must shift from chasing SOTA (State of the Art) benchmarks to _climate and resource awareness_. A researcher might develop a high-fidelity, multi-modal model for monsoon prediction. However, if this model requires massive resources, it contributes to the very climate instability that intensifies flooding in the Delta. True sustainability requires the community to institute **pre-registration and compute budgets**, forcing a pivot toward appropriate models that can be audited for their kgCO2e footprint._
>
> _The **industry and resource awareness** levels address the material "Base". Currently, large-scale forecasting is often a form of "data extractivism", where hydrological and weather data from the global South is ingested by corporations in global North to train proprietary models. To subvert this, industry must move beyond providing mere API access which creates a cycle of economic dependency and instead facilitate **local infrastructure ownership**. This could mean providing open-weight models and edge-compute hardware that allow local engineers to run inference without the resource overhead of multi-purpose, resource-heavy generative AI models. By grounding the "Base" in local control, we mitigate the social sustainability risk where the "solution" is controlled by a distant, profit-driven entity._
>
> _Finally, the **government and global action** levels address the "Superstructure". The regional governments cannot rely on the "goodwill" of tech giants; instead, it requires policy frameworks that mandate **data sovereignty** and transparent reporting. Global cooperation is the only mechanism to manage the "AI arms race" that currently drives the over-consumption of the global commons. International agreements must treat satellite weather data not as a licensed commodity, but as a public good. CARAML argues that managing climate change related risks with AI cannot be decoupled from the global imperative to reduce aggregate compute and redistribute the means of AI production and its benefits._
>
> ---
> ### Jevons Paradox in DeepSeek
> To better illustrate the rebound effect due to efficiency improvements and the related consumption surge, we provide additional evidence regarding DeepSeek model and the "reasoning model" trend. **We further refer to the response to Q1 of Reviewer FLFP.**
>
> ---
> ### Defining Sustainability
> We ground our work in the definition of sustainable AI based on [1] (L125-129):
> > how to develop AI that is compatible with sustaining environmental resources for current and future generations; economic models for societies; and societal values that are fundamental to a given society.
>
> We acknowledge that interpretations of sustainability can vary across stakeholders; however, this position specifically targets the ML community (research and academia). While the scope of our framework is broad, we prioritise the interpretation of these values as they apply to ML practitioners to ensure actionable insights within our field.
>
> ---
> ### Text Clarity and Additional References
> We are committed to sharpening the manuscript to eliminate redundancy. We thank the reviewer for the suggested references. The work by Shi et al. on regional disparities in GeoAI will strengthen our arguments regarding geographic inequity.
>
> ---
> ### References
>
>
> [1] Van Wynsberghe, A. Sustainable AI: AI for sustainability and the sustainability of AI. AI and Ethics, 2021.

---

> > ### Author Rebuttal · Reviewer_ubS3 · 2026-04-05
> >
> > Thanks for the rebuttal. My questions have been resolved.

---

### Decision · Program_Chairs · 2026-04-30

**Decision:**

Accept (regular)

**Comment:**

The paper presents a strong and convincing position that will add richness and nuance to the community’s discourse and future actions around sustainable AI. The paper presents convincing analysis and proposes a framework called Climate and Resource Aware Machine Learning (CARAML) to guide future discourse. In summary, there is a strong position, strong evidence, and a clear path forward.